# Comprehensive Environmental Assessment of Potato as Staple Food Policy in China

**DOI:** 10.3390/ijerph16152700

**Published:** 2019-07-29

**Authors:** Bing Gao, Wei Huang, Xiaobo Xue, Yuanchao Hu, Yunfeng Huang, Lan Wang, Shengping Ding, Shenghui Cui

**Affiliations:** 1Key Lab of Urban Environment and Health, Institute of Urban Environment, Chinese Academy of Sciences, Xiamen 361021, China; 2Xiamen Key Lab of Urban Metabolism, Xiamen 361021, China; 3University of Chinese Academy of Sciences, Beijing 100049, China; 4Department of Environmental Health Sciences, State University of New York at Albany, NY 12144, USA; 5School of Biotechnology Engineering, Jimei University, Xiamen 361021, China

**Keywords:** substitution ratio, chemical fertilizer inputs, irrigation-water use, total GHG emission, nutrient reference values

## Abstract

The Chinese government projected 30% of total consumed potatoes as a staple food (PSF) by 2020. We comprehensively assessed the potential impacts of PSF on rice and flour consumption, rice and wheat planting, energy and nutrient supply, irrigation-water, chemical nitrogen (N), phosphorus pentoxide (P_2_O_5_) and potassium oxide (K_2_O) fertilizer inputs and total greenhouse gases (GHG) emission for potatoes, rice and wheat, by assuming different proportions of potato substitutes for rice and flour. The results showed that per capita, 2.9 ± 0.3 and 4.7 ± 0.5 kg more potatoes per year would enter the Chinese staple-food diet, under the government’s target. PSF consumed are expected to reach 5.2 ± 0.7 Tg yr^−1^, equivalent to substituting potatoes for 4.2 ± 0.8–8.5 ± 0.8 Tg yr^−1^ wheat and 5.1 ± 0.9–10.1 ± 1.8 Tg yr^−1^ rice under different scenarios. While this substitution can increase the nutrient supply index by 63% towards nutrient reference values, it may induce no significant effect on staple-food energy supply with lower chemical fertilizer (except for K_2_O) and irrigation-water inputs and GHG emissions in different substitution scenarios than the business as usual scenario. The reduction in rice and wheat demands lead to wheat in the North China Plain and early rice decrease by 6.1–11.4% and 12.1–24.1%, respectively. The total GHG reduction is equal to 1.1–9.0% of CO_2_ equivalent associated with CH_4_ and N_2_O emitted from the Chinese agroecosystem in 2005. The saved irrigation water for three crops compared to 2012 reaches the total water use of 17.9 ± 4.9–21.8 ± 5.9 million people in 2015. More N fertilizer, irrigation-water, and GHG can be reduced, if the PSF ratio is increased to 50% together with potato yield improves to the optimal level. Our results implied that the PSF policy is worth doing not only because of the healthier diets, but also to mitigate resource inputs and GHG emissions and it also supports agricultural structure adjustments in the areas of irrigated wheat on the North China Plain and early rice across China, designed to increase the adaptability to climate change.

## 1. Introduction

Potatoes rank as the fourth largest food crop after rice, wheat and corn in the world, as well as in China. With a 400-year history of potato cultivation, China is now the world’s largest potato producer. Planting area and yield reached 5.5 million ha and 19.0 million tons in 2013 [1], yielding approximately 20% of the world output [2]. However Chinese per capita consumption of potatoes was only 41 kg in 2011, ranking 65th in the world, far less than the per capita average of 84 kg in Europe and 185 kg in Belarus; the latter figure is equivalent to a Chinese resident’s total annual consumption of wheat-flour staple foods [3]. In the EU and North America where one to two thirds of the daily potato consumption is in a processed form such as French fries and potato chips [4]. However the Chinese per capita annual consumptions of potato flour and potato chips are only about 73 and 15 g, contributing to 0.4% and 0.2% of the total per capita potato consumption, respectively [5]. In other words, potatoes are mostly consumed as fresh vegetables in the current Chinese diet.

The Chinese government decided to promote potatoes as a staple food (PSF) in 2015, and set a target of 30% of the total potatoes consumed as a staple-food by 2020 [1], based on the following reasons, first, to meet the public’s demand for tasty and healthful food [5], and the dietary needs of the fast-paced 21st-century lifestyle of both urban and rural residents [6]. Second, potatoes can become a major alternative crop in the agricultural structural adjustment that is needed to respond to limited arable land and declining water availability, because they can be grown with less water, land and fertilizer than cereal crops. Most importantly, with population and economic growth, Chinese grain demand is expected to increase by 6.9% and 3.3% for rice and wheat by 2030, respectively, relative to 2012 [7]. Due to the limitation on arable land expansion in China [8], producing more food might occur at the expense of increasing nutrient inputs if no other improvements in agronomic management are made [7]. This presents the challenge of producing more grains with fewer inputs while reducing environmental costs. Potato as a staple food could meet 500 million tons of the food needs anticipated by China’s increasing population, if the planting acreage could reach 150 million mu (1 mu = 667 m^2^) by 2020 [6]. The governments’ goal is to process potatoes for adapting to the Chinese consumer’s eating habits for staple foods such as bread, steamed bread, and noodles, thereby changing potatoes from a non-staple food to a staple one [1].

Recent literatures shed light on the understanding of the nexus of food–energy–water for tackling interdependencies among food, energy, and water security [9,10]; reducing greenhouse gas (GHG) emissions, risks of eutrophication and land use demand by dietary recommendations [11]; Revealing the multiple environmental benefits of animal food consumption by calculating the land–water–GHG–N burdens of per consumed calorie of various livestock categories, and presenting recommendations for reducing multiple environmental benefits by guiding dietary changes to lower consumption of animal-derived food and reduce consumption of the uniquely high resource demands of beef [12]. These findings highlight that the changes in final food consumption—the demand for food, diet changes and selective food consumption—may drive the changes in land use, water, energy, GHG, and N inputs or burdens.

Developing potatoes as a staple food would decrease the consumption of flour and rice. The resulting reduction in rice and flour would lead to less wheat and rice planting—one of the goals of the agricultural structural adjustment in some regions of China. Potatoes have different chemical nitrogen (N), phosphorus pentoxide (P_2_O_5_) and potassium oxide (K_2_O) fertilizer and irrigation-water inputs and use efficiencies, and significant lower GHG emissions, compared with rice and wheat (see Appendix A for details and Appendix A). Many studies were conducted, mainly focusing on the current status of the consumption of potato and its related products in Chinese residents during 2010–2012 [5], a certain amount of potato flour addition could improve the quality of wheat flour and noodle [13], potatoes steamed bread as staple foods improved the nutrition structure of a Chinese resident by significantly enhancing vitamin C, potassium, protein and dietary fiber [3], and GHG emissions, expressed in the form of a CO_2_ equivalent (CO_2_-eq) from agricultural inputs in potatoes production are significantly lower than that in wheat and maize cultivation [14]. However, the shortage of integrated research remains, in particularly in quantifying the potential impacts of PSF on rice and flour consumption, rice and wheat production, resources input and total GHG emissions for rice, wheat and potatoes production behind the changes in staple food structure when potatoes entered into staple diets in China. The comprehensive evaluation of the impacts of potato staple food policies on multiple resource inputs, environmental indicators and food energy supplies are paramount for food security, cleaner production and sustainability of food production in China.

This study aimed (1) to analyze the future trends in per capita staple-food and potato consumptions in 2020 based on the historical trends from 1980 to 2012; (2) to evaluate the comprehensive effects of PSF on the nexus of staple food consumption and production, chemical fertilizer and irrigation water inputs, total GHG emissions, food energy supply, nutrition structure, by assuming different proportions of potato substitutes for rice and wheat flour; (3) to assess the spatial patterns of potato systems substitute for winter wheat in the North China Plain (NCP) and early rice across China. These findings will help to explore solutions to produce more staple food with lower inputs and with reduced environmental costs in China and may provide a reference template for sustainable water management studies in the areas with serious water deficit.

## 2. Methodology

### 2.1. Data Collection

The datasets about urban and rural population, the selected cropping systems sowing area, per capita habitual staple food consumption were mainly taken from China's statistical yearbooks and bulletins and the Ministry of Health in China [15,16,17]. The second category of data is coefficients used for the calculation of the national chemical fertilizer inputs, irrigation-water consumption and total GHG emissions, e.g., the per hectare chemical N-, P_2_O_5_- and K_2_O-fertilizer and irrigation-water inputs, yield and their use efficiencies, power used per unit of irrigation rate and GHG emissions from different sources in early rice, medium rice, late rice, winter wheat in the NCP and winter wheat across China except for NCP and potato production (Appendix A), were collected from the published literature between 2000–2017. The principle for collecting these data was that at least one of the indicators—such as chemical N-, P_2_O_5_- and K_2_O-fertilizers, or irrigation-water input for the three crops—was reported. In total, 508 results were collected, which were divided between each of the cropping systems as follows; conventional potatoes (111) optimized potatoes (109), early rice (68), medium rice (44), late rice (48), winter wheat in the NCP (86) and winter wheat across China except for NCP (42; Appendix A). The ratio of organic fertilizers’ application area to total sown area is relatively low, especially in rice and wheat production, at the same time, the amounts of N, P_2_O_5_ and K_2_O from organic fertilizer were not directly reported in the selected literature as that of chemical fertilizers, and they can not be easy to calculate because the complex types of organic fertilizers and no available data on the contents of N, P_2_O_5_ and K_2_O in some organic fertilizers. In addition, the organic N-, P_2_O_5_- and K_2_O-fertilizers belong to the internal recycled resources in the food production–consumption system [18,19], and we mainly emphasize the new resources input into the food system and evaluate the contribution of PSF to the national goal of zero growth in fertilizer use, hence, the N, P_2_O_5_ and K_2_O sourced from organic fertilizers were excluded in this study. The mean partial factor productivities of chemical N, P_2_O_5_, K_2_O (PFP_N_, PFP_P2O5_ and PFP_K2O_, in kilograms of standard grain per kilogram of N, P_2_O_5_ and K_2_O applied) and irrigation-water use efficiencies (IWUE) for each crop were estimated by the yield divided by the relevant inputs (Appendix A). In addition, we calculated the mean irrigation rate for rice across China using early, medium and late rice irrigation amounts multiplied by each sown area, then divided by the total sown area for rice (Appendix A).

GHG emissions from rice, wheat and potatoes production include soil N_2_O and CH_4_ emissions, the indirect CO_2_ from the manufacture and transportation of the chemical fertilizer, power use for irrigation, fuel combustion in farm operations, application of pesticide and film for mulching crops [20,21]. The detailed principle of collecting the above GHG emissions was described in SI Text. In total, 148 results of N_2_O and CH_4_ emissions were collected, which were divided between each of the cropping systems as follows; early rice (26), medium rice (22), late rice (41), wheat in the NCP (29), wheat across China except for NCP (24) and potatoes (6; Appendix A). We calculated the total GHG (kg CO_2_-eq ha^−1^) of different crops by summing all GHG emissions from soil and agronomy management, then calculated yield-scale GHG intensity (GHGI, kg CO_2_-eq kg^−1^ grain) [20], for potatoes, winter wheat in the NCP and early rice across China.

### 2.2. Changes in Per Capita Staple-Food and Potato Consumption in China, 1980–2012, and Future Trends

We collected per capita habitual staple-food (including rice, flour, other cereals, tubers and beans) intake data for Chinese urban and rural residents in 1982, 1992, 2002 and 2012 from the Ministry of Health in China [15,16] (Appendix A). We estimated per capita habitual food intake for the non-sampling years during the period 1980–2012, by linear interpolation between every two adjacent intervals of the surveys [22], and extrapolated per capita habitual staple-food intake for 2013–2020 based on the historical trends of each staple-food intake from 2002 to 2012, using a simply common trend projection method [23]. We then estimated per capita food consumption by urban and rural residents combined with the ratio of kitchen wastes (Appendix A).

In 2012, per capita potato consumption values were 7.7 and 14.5 kg yr^−1^ for urban and rural residents, respectively, accounting for 70% and 88% of per capita tuber (including mainly potatoes, sweet potatoes, yams and tania) consumption in urban and rural areas; however, as mentioned in introduction potatoes are mostly consumed as fresh vegetables in the current Chinese diet (Fang et al., 2016). We predicted the amounts of potatoes consumed as a vegetable in urban and rural populations in 2020 by historical trends of tuber consumption from 2002 to 2012, and assuming that the proportion of potatoes to other tubers keep constant as in 2012. Then the PSF consumption calculated with the following equation:SP = VP × R_PSF_/(1 − R_PSF_),(1)
where the *VP* is potato-as-vegetable consumption amounts, the *R_PSF_* is the government’s target of 30% of the total potatoes consumed as a staple-food, and we added 50% as an additional potential target (Table 1). The *VP* was calculated as 6.7 ± 1.0 and 11.0 ± 1.7 kg yr^−1^ for urban and rural residents, respectively, in 2020.

### 2.3. Scenario Analysis

For the business as usual (BAU) scenario, we assumed a Chinese population of 1.44 billion people with an urbanization level of 60% in 2020 [24,25], and that per capita rice, flour and potato-as-vegetable consumptions changed at the same rate as historical trends during 2002–2012; the PSF consumption had no effect on the rice-or flour-as-staple-food consumptions; the production efficiencies of chemical N-, P_2_O_5_- and K_2_O-fertilizers and irrigation water, and GHG emissions of rice, wheat and potatoes were the mean levels of the collected data in this study (Appendix A), which can represent the mean pattern of farmers’ practices in the last three decades and the near future as much as possible, hence we assumed that these efficiencies and GHGI of three crops no variation in all scenarios, and potato planting reached 6.7 million ha that has been proposed by the government [1]. For PSF scenarios, we adopted the Chinese government’s target of having 30% of total consumed potatoes as a staple food by 2020, and set a more ambitious target of increasing the proportion of PSF to 50%. We further divided several sub-scenarios by assuming different proportions of potato substitutes for rice and flour (Table 1). Amounts of per capita consumption of PSF, under these different scenarios, were calculated according to the PSF consumption ratio and the per capita potato-as-vegetable consumption ratio in 2020 (Table 2). We then estimated the national PSF consumption under the different scenarios, by multiplying the population of rural and urban residents with the per capita PSF consumption in 2020, and estimated the consumptions of rice and flour, and the demand for rice and wheat grain, fertilizer and irrigation-water inputs and GHG emissions for three crops under the different scenarios (Appendix A), based on an equivalent replacement for rice and flour using potato powder [1,3]. The detailed information on the calculations of the demand for rice and wheat grain, fertilizer and irrigation-water inputs and GHG emissions under different scenarios are described in the SI text. We further compared the total energy supply between different potato substitutions for rice and flour scenarios and only rice and flour scenario in 2012, expressed as calorie (Appendix A); and compared the nutrient supply index between 35% potato bread (250 g) and 100% wheat bread (250 g), which was calculated by the ratio between the provide nutrition from 35% potato bread (250 g) and 100% wheat bread (250 g) and nutrient reference values (NRV) given by the association of China dietary nutrition, multiply by the average weight of 14 nutrient index (includes crude protein, crude fiber, Vitamin B_1_, Vitamin B_2_, Vitamin C, potassium (K), phosphorus (P), magnesium (Mg), calcium (Ca), (tellurium) Te, zinc (Zn), manganese (Mn), copper (Cu) and selenium (Se); Appendix A). The nutrient supply index can represent the extent of each nutrient supply close to the NRV, and we used the sum of these indices to evaluate the health effect of substitution PSF for rice and flour.

### 2.4. Effect of PSF on Rice and Wheat Production in Different Scenarios

The spatial patterns of early rice, winter wheat and potato systems at the county-level (Appendix A), were developed from a 30 m × 30 m resolution land use map provided by Wu et al. [26] (see Appendix A for details). Winter wheat in the NCP was further divided into irrigated- (NCP-I) and rain-fed winter wheat (NCP-R) depended on their irrigation conditions [27]. Substituting rice and flour with potatoes as staple foods means we could cut down rice and wheat planting in China. In this study, we assumed rice reduction only in early rice planting, because it has lower yield and quality compared to medium and late rice. We also assumed that the decrease in winter wheat sowing would occur in the NCP, because this region is facing rapid depletion of groundwater due to the planting of winter wheat needs for excessive ground water for irrigation [28,29]. The reductions of early rice and winter wheat in the NCP were calculated by the substituting rice and wheat grain with potatoes, divided by per unit area yield of early rice across China and winter wheat in the NCP.

### 2.5. Variability and Uncertainty Analysis

There are uncertainties in estimating different staple food consumption, national chemical N-, P_2_O_5_-, K_2_O-fertilizer and irrigation water inputs, total GHG emissions, etc. driven by the variability of the collected coefficient data on per hectare fertilizer and irrigation inputs, all GHG emission from soil and agronomy management, and activity data from China’s Statistical Yearbook and the Ministry of Health in China, such as crop sowing area, per capita habitual staple food consumption and human population, urbanization rate. We calculated the mean and variation with the 90th percentile confidence interval of the collected data on per hectare fertilizer and irrigation inputs, all GHG emission from soil and agronomy management, as cited in Appendix A, and assumed that uncertainties in results driven from activity data are within a range of ∼5% [19,30], such as rice, wheat and potato sown area, human population and urbanization rate. The per capita potatoes consumption [5], the per capita habitual staple food consumption from the Ministry of Health in China, the predicted the amounts of tuber consumed as a vegetable in urban and rural populations in 2020 by historical trends of tuber consumption from 2002 to 2012, per capita water consumption in 2015 [31], and the predicted population of 2020 given by Wang et al. [24], and we assumed that these data will drive 10% uncertainties for our results. An uncertainty analysis was performed using the error propagation equation of mathematical statistics [32] (see Appendix A for details). The means and likely ranges are reported in associate contents, figures and tables.

## 3. Results

### 3.1. Variations in Staple Food Consumption Per Capita and Its Composition

Per capita staple-food consumption, and the contributions of each food type to the total staple-food consumption in Chinese urban and rural households, dramatically changed during 1980–2012 (Figure 1). In the urban setting, total staple-food consumption per capita fell steadily from 223.6 ± 13.1 to 131.0 ± 7.7 kg cap.^−1^ yr^−1^, but the decrease rate slowed after the year 2002, because small increases in the consumptions of flour and other cereals offset the decreases in the consumption of rice, tubers and beans. The consumption of tubers, including mainly potatoes, sweet potatoes, yams and tania, showed a decrease from 27.0 to 10.9 kg cap.^−1^ yr^−1^. Rice and flour together accounted for more than 80% of total staple food consumption, followed by potatoes, other cereals, beans and tubers (excluding potatoes), whose shares were 6%, 5%, 4% and 3% in 2012, respectively. In the rural setting, total staple-food consumption per capita has followed the same trend as in the urban setting, decreasing from 329.5 ± 16.5 to 183.1 ± 11.5 kg cap.^−1^ yr^−1^; rice showed an increase in the 1980s, and then fell to about 99.0 kg cap.^−1^ yr^−1^ in the 2010s. Flour showed the same trend as rice, maintaining at roughly 55.0 kg cap.^−1^ yr^−1^ in the 2010s. As for other cereals and tubers, they clearly decreased, from 70.3 and 97.7 kg cap.^−1^ yr^−1^ in 1980 to 7.8 and 16.4 kg cap.^−1^ yr^−1^ in 2012, respectively. The proportions of rice accounted for 28–54% of staple food consumption during 1980 to 2012, while flour increased from 19% to 30%, and other cereals and tubers showed a dramatic reduction, from 21% and 30% to 5% and 9%, respectively. The per capita rice and flour consumption amounts were predicted to reach 50.7 and 49.8 kg yr^−1^ for urban residents, and 98.1 and 55.6 kg yr^−1^ for rural residents, respectively, in 2020, based on their trends during 2002–2012. Together, they accounted for 82% and 88%, respectively, of the per capita total staple-food consumption in urban and rural areas.

If the targets of 30% and 50% of the total PSF consumption (additional annual per capita consumptions of more than 2.9 ± 0.4 and 6.7 ± 1.0 kg yr^−1^ for urban areas, and 4.7 ± 0.7 and 11.0 ± 1.7 kg yr^−1^ for rural areas, respectively) are achieved, the potato will enter the Chinese diet as a staple food by 2020 (Table 2). The amounts of PSF currently account for 2.6% and 6.7%, respectively, of the weight of rice and flour consumed in urban areas, and 3.1% and 7.2%, respectively, in rural areas.

### 3.2. Effect of PSF on Potatoes, Rice and Flour Consumption, and on Potatoes, Rice and Wheat Production under Different Scenarios

In 2012, the national rice, flour and potatoes consumption was 189.5 ± 12.6 Tg, including 105.1 ± 10.5 Tg rice, 70.3 ± 7.0 Tg flour and 14.1 ± 0.7 Tg potatoes (Figure 2). Under the BAU, the total consumption will increase to 192.4 ± 19.8 Tg by 2020. However, rice consumption will decrease by 5.0 ± 0.5 Tg yr^−1^, flour and potatoes consumption will increase by 4.6 ± 0.4 and 3.3 ± 0.1 Tg yr^−1^, respectively, compared with 2012, and an additional 5.2 ± 0.7 Tg yr^−1^ potato consumption will enter household diets as staple food. Under the 30% scenarios, this 5.2 ± 0.7 Tg PSF can substitute 2.6 ± 0.4–5.2 ± 0.7 Tg of rice and flour, respectively, under the scenarios of 30S_0R+100F_, 30S_50R+50F_ and 30S_100R+0F_. Rice consumption would be maintained at the same level as under the BAU if PSF completely replaced flour under the 30S_0R+100F_ scenario, while it would decrease by 2.6% for 30S_50R+50F_ and 5.2% for 30S_100R+0F_ relative to the BAU. Flour consumption would be maintained at the same level as under the BAU if PSF completely substituted for rice under the 30S_100R+0F_ scenario, while it would decrease by 3.5% for 30S_50R+50F_ and 6.9% for 30S_0R+100F_, relative to the BAU. Under the 50% scenarios, an additional 7.0 ± 0.4 Tg yr^−1^ potato consumption would enter the household diet as a staple food relative to 30% scenarios, and could substitute for 6.1 ± 0.9 and 12.2 ± 1.7 Tg rice or flour yr^−1^, respectively, under the scenarios of 50S_0R+100F_, 50S_50R+50F_ and 50S_100R+0F_. Rice consumption would decrease by 6.1% for 50S_50R+50F_ and 12.2% for 50S_100R+0F_ relative to the BAU, and flour consumption would decrease by 16.3% for 50S_0R+100F_ and 8.1% for 50S_50R+50F_ relative to the BAU. However, food energy supply in different potatoes substitution scenarios only increase by 0.4–1.6% compared to the only rice and flour scenario in 2012.

The potential effect of PSF on rice and wheat production was also estimated (Figure 3), by considering the relationships between initial grain production and the final rice and flour consumption by households. The results showed that the demand for rice would decrease by 4.6% under the BAU scenario, but the demand for wheat would increase by 6.6% over 2012. Under the 30S_0R+100F_ and 50S_0R+100F_ scenarios, PSF could substitute for 8.5 ± 1.5 and 19.8 ± 3.6 Tg wheat yr^−1^, respectively, accounting for 7.0% and 16.3% of China’s wheat demand in 2020, under the BAU. Under the 30S_100R+0F_ and 50S_100R+0F_ scenarios, potatoes could substitute for 10.1 ± 1.8 and 23.7 ± 4.3 Tg rice grain yr^−1^, accounting for 5.2% and 12.2% of the rice demand in 2020 under the BAU. Under the 30S_50R+50F_ and 50S_50R+50F_ scenarios, potatoes could substitute for 4.2 ± 0.8 and 9.9 ± 1.8 Tg wheat yr^−1^, and 5.1 ± 0.9 and 11.8 ± 2.1 Tg rice yr^−1^, respectively. Substituting rice and winter wheat with potatoes may lead to land use change, such as converting early rice across China or irrigated winter wheat in the NCP to potato cultivation (Appendix A). Under the 30S_0R+100F_ and 50S_0R+100F_ scenarios, PSF could substitute 1.3 ± 0.4 × 10^6^ and 3.0 ± 0.9 × 10^6^ ha winter wheat sowing in the NCP, or 1.4 ± 0.4 × 10^6^ and 3.3 ± 0.9 × 10^6^ ha early rice could be substituted by potatoes in 30S_100R+0F_ and 50S_100R+0F_ scenarios. Under the 30S_50R+50F_ and 50S_50R+50F_ scenarios, potatoes could substitute for 0.7 ± 0.2 × 10^6^ and 1.5 ± 0.4 × 10^6^ ha winter wheat in the NCP, and 0.7 ± 0.1 × 10^6^ and 1.7 ± 0.3 × 10^6^ ha early rice across China, respectively.

### 3.3. GHG Emissions and GHGI for Producing Potatoes, Wheat and Rice

The GHG emissions for producing potatoes in conventional and optimized practices, winter wheat in the NCP, winter wheat across China except for NCP, early rice, medium rice and late rice was 3636 ± 1140, 3323 ± 1037, 4677 ± 2329, 4212 ± 1360, 9138 ± 5059, 9943 ± 3993 and 11,393 ± 3993 kg CO_2_-eq ha^−1^, respectively (Figure 3). GHGI for producing potatoes in conventional and optimized practices, winter wheat in the NCP and early rice was 0.7 ± 0.3, 0.5 ± 0.2, 0.7 ± 0.4 and 1.3 ± 0.7 kg CO_2_-eq kg^−1^, respectively. This result explained that the total GHG for producing staple food reduced when substituting rice and flour with potatoes.

### 3.4. Total Chemical N-, P_2_O_5_- and K_2_O-fertilizer, Irrigation-Water Consumption and total GHG Emissions for Producing Potatoes, Wheat and Rice

Total chemical N-, P_2_O_5_- and K_2_O-fertilizer applications and irrigation-water used for the three crops were 11.9 ± 3.7, 5.6 ± 0.6 and 5.7 ± 0.9 Tg yr^−1^ and 239.6 ± 82.5 × 10^9^ m^3^ yr^−1^ under conventional practices in 2012 (Appendix A). Under the BAU scenario, chemical N, P_2_O_5_ and K_2_O applications had smaller increases—by 0.2 ± 0.05 Tg yr^−1^—while irrigation-water decreased by 5.9 ± 1.5 × 10^9^ m^3^ yr^−1^, under 2012 amounts (Figure 4). PSF scenarios have different effects on total chemical N, P_2_O_5_ and K_2_O applications and irrigation-water use, when using potato substitutes for different proportions of rice and flour, because potatoes have different chemical fertilizers and irrigation-water inputs and use efficiencies with early rice across China and winter wheat in the NCP (Appendix A). Compared with BAU scenario increased chemical N, P_2_O_5_ and K_2_O applications for three crops under conventional practices, all the substitute scenarios mitigate N, P_2_O_5_ and K_2_O applications, except for K_2_O applications in three 50% of potatoes as staple food scenarios, and irrigation water and GHG emissions further decreased. The total GHG was 427.1 ± 96.3 Gg CO_2_-eq for producing potatoes, rice and winter wheat in 2012, and it decreased by 3.4 ± 0.6 Gg CO_2_-eq in BAU scenario because of the reduction in rice consumption in 2020 relative to 2012. Under different substitution scenarios, total GHG can reduce by 9.4 ± 1.7–23.3 ± 3.8 Gg CO_2_-eq under conventionally grown potatoes and 14.1 ± 2.5–32.6 ± 2.5 Gg CO_2_-eq under optimally grown potatoes, compared to 2012. The highest reduction appeared in 50S_100R+0F_ scenario because of the higher GHGI of early rice than potatoes and winter wheat (Figure 3).

Potato yield in China was found to be 25.8 Mg ha^−1^, using conventional practices: 25.6% lower than if optimized practices were used. If the potato yields could be increased from the current conventional level to an optimized practices level, total chemical N, P_2_O_5_ and irrigation-water use for three crops would be further decreased, compared to 2012, because of the higher two fertilizer use efficiencies and IWUE in optimized practices than in conventional practices (Appendix A). However, K_2_O application will slightly increase because potatoes like K_2_O more than rice and wheat.

## 4. Discussions

### 4.1. Effect of PSF on Rice and Flour Consumption and Rice and Wheat Production

The per capita total consumption of potatoes as a staple food and as a vegetable, in 30% and 50% of the PSF scenarios, would be 9.6 ± 1.1 and 13.5 ± 1.4 kg yr^−1^ in urban areas, and 15.7 ± 1.8 and 22.1 ± 2.3 kg yr^−1^ in rural areas, respectively, in 2020 (Table 2). Even these higher amounts, though, are lower than the world average annual per capita potatoes consumption of 20.8 kg yr^−1^ (except for the 50% PSF scenario in rural areas), and far below the potato consumption per capita in 2005 in Europe (96.1 kg) and North America (57.9 kg) [4]. Asia as a whole, though, is an area with the fastest growth of potato consumption, and per capita potato consumption amounts were as large as 40–80 kg yr^−1^ in some countries, such as Kazakhstan, Lebanon, Israel, etc. [33]. Hence China has a large potential for growth of potatoes as a staple food, and the Chinese government’s goal will increase potatoes as a staple food amounts to 5.2 ± 0.7 and 12.2 ± 1.7 Tg yr^−1^ under 30% and 50% of PSF targets. Such an increase can substitute for 3.1% and 7.5% of the total rice and flour consumption in 2020, would reduce flour consumption by 5.2 ± 0.7 and 12.2 ± 1.7 Tg yr^−1^ under the 30S_0R+100F_ and 50S_0R+100F_ scenarios; or reduce rice consumption by almost identical amounts under the 30S_100R+0F_ and 50S_100R+0F_ scenarios; or decrease rice and flour consumption amounts by 2.6 ± 0.4 and 6.1 ± 0.9 Tg yr^−1^, respectively, under the 30S_50R+50F_ and 50S_50R+50F_ scenarios.

The ratios of grain supplies for the different types of flour in China are about 78.0%, and about 3.0% of grains are lost during processing [34]. Under the scenarios where potatoes are completely substituted for flour as a staple food (30S_0R+100F_, 50S_0R+100F_), and those where potatoes are substituted for half the flour (30S_50R+50F_, 50S_50R+50F_), 8.5 ± 1.5, 19.8 ± 3.6, 4.2 ± 0.8 and 9.9 ± 1.8 Tg wheat yr^−1^ or 7.0%, 16.2%, 3.5% and 8.1% of the total wheat demand (121.9 ± 23.6 Tg yr^−1^) under the BAU scenario will be substituted, respectively, in 2020. The edible portion of rice is 70.5%, and about 3.0% of rice is lost in processing [34]. Under the scenarios where potatoes are completely substituted for rice as a staple food (30S_100R+0F_, 50S_100R+0F_) and those where potatoes are substituted for half the rice as a staple food (30S_50R+50F_, 50S_50R+50F_), 10.1 ± 1.8, 23.7 ± 4.3, 5.1 ± 0.9 and 11.8 ± 2.1 Tg yr^−1^ or 5.2%, 12.2%, 2.6% and 6.1% of the total rice demand (194.6 ± 37.8 Tg yr^−1^) under the BAU scenario will be substituted in 2020.

With population and economic growth, the demand for grain in China is expected to reach 218 Tg for rice and 125 Tg for wheat by 2030, compared to the 204 and 114 Tg of rice and wheat demands in 2012 [7]. The results of the current study indicate that the demands for rice for staple-food consumption will fall to about 195 ± 38 Tg by 2020, because of the decline in per capita rice consumption, especially in urban resident’s diets (Figure 1). Indeed, the rice reduction has already exceeded the expected increase in rice demand from population growth [22]. But the demand for winter wheat is expected to reach 122 ± 24 Tg by 2020, according to a trend found in a previous study, because per capita flour consumption has been relatively stable in recent decades, keeping pace with population growth. However, Chinese agriculture is restricted by limited arable land, declining water availability, increasing costs for rural labor, and increasing vulnerability to climate change [35], resulting in difficulty in increasing cereal production. The increased demand for wheat will be only relying on improving yields. It was indicated that, using the same planting area as in 2012, the total production of wheat would reach 174 Tg by 2030 if wheat grain yields of farmers could reach to 80% of the attainable yield level through integrated soil–crop management systems [7].

PSF could reduce the grain demand for rice and/or winter wheat under the different substitution scenarios. For example, under the 30S_50R+50F_, 30S_0R+100F_, 50S_50R+50F_ and 50S_0R+100F_ scenarios, the demands for rice could be expected to reach only 190 ± 36, 185 ± 34, 183 ± 31 and 171 ± 31 Tg—lower by 2.6%, 5.1%, 6.0% and 12.1% than that under the BAU scenario, by 2020. Under the scenarios of 30S_0R+100F_, 30S_50R+50F_, 50S_0R+100F_ and 50S_50R+50F_, the demands for wheat could be expected to reach only 113 ± 21, 118 ± 22, 102 ± 18 and 112 ± 20 Tg—all lower than that under the BAU scenario, by 2020, close to the actual demand in 2012. These results imply that if China could achieve the targets of 30–50% PSF consumption, the wheat production likely would meet the demand placed on it by population growth by 2020, under current yields and arable land availability across China. Additionally, the planting area required for wheat production could be reduced by 28% relative to wheat planting area in 2012, if farmers could achieve wheat grain yields of 80% of the yield level attainable through integrated soil–crop management systems [7,36].

### 4.2. Effect of PSF on Chemical N-, P_2_O_5_- and K_2_O-fertilizer Application on Rice, Wheat and Potato Systems

Different substitute amounts of potatoes for rice and wheat, integrated with the use efficiencies of chemical N-, P_2_O_5_- and K_2_O-fertilizer and irrigation-water in different cropping systems, will generate certain effects on total chemical N-, P_2_O_5_-, K_2_O-fertilizer and irrigation-water consumption for these crop systems (Figure 4): N application on the three crops would fall slightly, by 0.2 ± 0.1–0.4 ± 0.1 Tg yr^−1^ relative to the BAU, and we found that the more the amount of wheat substituted by potatoes, the more the N would be reduced, because the PFP_N_ for potatoes is higher than that for wheat (35.8 vs. 28.2 kg kg^−1^ N). Substituting potatoes for rice has the opposite effect: The P_2_O_5_ amounts for the three crops would be slightly reduced, by 0.1 ± 0.01–0.2 ± 0.02 Tg yr^−1^, the more potatoes that are substituted for rice, the less P_2_O_5_ can be decreased relative to the BAU, because the PFP_P2O5_ for rice is higher than that for potatoes (98.8 vs. 57.3 kg kg^−1^ P_2_O_5_). K_2_O application on three crops would decrease slightly, by 0.1 ± 0.02–0.2 ± 0.03 Tg yr^−1^ in three 30% PSF scenarios relative to the BAU, but showed a slight increase in three 50% PSF scenarios, because the lower PFP_K2O_ for potatoes than that for rice and wheat (39.3 vs. 68.7 and 81.3 kg kg^−1^ K_2_O). China’s conventional potato yield is higher than the world average by about 25.4%, however, a relative large potential for improving potato yields by 25.6% (up to 32.4 t ha^−1^) through optimizing management practices. The optimized potato yield is only about 70% of that in the USA, Germany, France and the Netherlands, even though China is among the top 10 countries for potato output (Appendix A) [2]. If the potato yield could be improved to the optimized level, further more reductions of N, P_2_O_5_ and K_2_O could be achieved, relative to the BAU, under all the substitution scenarios, except for K_2_O in three 50% PSF scenarios.

In the past decade, the mean increase rates of chemical N-, P_2_O_5_- and K_2_O-fertilizer application were 0.49, 0.43 and 0.41 Tg yr^−1^, respectively [17]. About 34.5% of the N, 32.4% of the P_2_O_5_ and 34.6% of the K_2_O were consumed by rice, wheat and tuber production in China [37]. The chemical N, P_2_O_5_ and K_2_O applications on rice, wheat and potatoes will slightly increase under the BAU scenario. However, chemical N-, P_2_O_5_- and K_2_O-fertilizer inputs to rice, wheat and potatoes showed different levels of reduction in all of the substitution scenarios except for K_2_O under three 50% PSF scenarios, relative to BAU. The latest statistics suggested that the agricultural fertilizer use decreased by 0.4 Tg in 2016 compared to 2015 [17]. This is the first negative growth since the 1970s and achieves the national goal of zero growth in fertilizer use three years in advance [30], which is attributed to the long-term effort to implement the soil testing and fertilizer recommendation technology from 2005, and the increase in crop yield and the ratios of straw return to field in majority of provinces [36,38], and the increase in manure caused by increases in livestock and poultry production [39]. Substituting rice and flour with PSF will lead to additional decrease in chemical N-, P_2_O_5_- and K_2_O-fertilizer inputs for the increasing food production in China from a long-term perspective.

### 4.3. Effect of PSF on Irrigation-Water Use and Total GHG for Rice, Wheat and Potatoes, and for Rice and Wheat Production

PSF can also decrease irrigation-water use for three crops, and we found that the greater amount of potatoes substituted for rice, the greater amount of irrigation-water would be reduced (Figure 4), because rice has a low IWUE relative to potatoes and wheat (Appendix A). The saved irrigation-water compared to 2012 by substituting rice and flour in varying proportions of potatoes, as a staple food, is equal to the total water use of approximately 17.9 ± 4.9, 20.0 ± 5.4 and 21.8 ± 5.9 million people (based on per capita use of 445 m^3^) in 2015 [31], in the 30% of PSF scenarios. Under the 50% of PSF scenarios, these amounts come to the total water use of approximately 18.2 ± 4.9, 22.5 ± 6.0 and 26.7 ± 7.1 million people in 2015. These reductions could alleviate the shortage of freshwater resources in China. If potato yields could be improved to the optimized level, even more irrigation-water could be saved, under all the substitution scenarios. The largest reduction in irrigation-water was found for the 50S_100R+0F_ scenario, equal to the total water use of about 35.1 ± 7.1 million people in 2015.

Wheat yield would be reduced by 8.5 ± 1.5, 4.2 ± 0.8, 19.8 ± 3.6 and 9.9 ± 1.8 Tg under the 30S_0R+100F_, 30S_50R+50F_, 50S_0R+100F_ and 50S_50R+50F_ scenarios, relative to the BAU, by 2020. These amounts are equal to about 1.3 ± 0.4, 0.7 ± 0.2, 3.0 ± 0.9 and 1.5 ± 0.4 million ha for winter wheat in the NCP, based on a yield of 6.6 Mg ha^−1^ of winter wheat in the NCP in 2012, accounting for 11.4%, 6.1%, 26.3% and 13.2% of the total winter wheat sown (11.4 million ha) in the NCP [27]. The possible region for cutting winter wheat by recommending PSF locate at NCP-I in Appendix A, where it is facing a rapid depletion of groundwater due to excessive irrigation for winter wheat [28,40]. Some recent studies have been carried out, on the possibility of saving irrigation water by converting to a winter wheat–summer maize–spring maize crops system, over a two-year period, rather than the typical winter wheat–summer maize double-cropping system, in the NCP [41,42]. The decrease in winter wheat production could extend the spring maize–summer maize double cropping system in the NCP, which in turn could decrease irrigation water by 28.0–62.7%, while increasing maize yield by 43.7–58.6% compared to the conventional winter wheat–summer maize double-cropping system [43,44]. This change fits in well with the trend of increasing the amount of maize for animal feed, as the demand for animal protein increases and staple foods such as wheat are decreasing in the human diet as economic development proceeds [22]. If potatoes can substitute for winter wheat, irrigation water requirements would decrease from 182 mm to 97 mm, and adding in the mean annual 70 mm irrigation for summer maize [42] would reduce the irrigation water requirement from 252 mm for the winter wheat–summer maize double cropping system to 167 mm for the potatoes and maize double cropping system. It has been found that groundwater exploitation should be controlled to below 150 mm yr^−1^ to slow the declining in groundwater table in the NCP [45]. Sowing potatoes therefore has a larger potential for mitigating the drop in the groundwater table than does sowing wheat, and it might be a good adaption to future climate change, with its expected significant warming trend and lower precipitation amounts in the NCP [46].

Rice yield would be reduced by 5.1 ± 0.9, 10.1 ± 1.8, 11.8 ± 2.1 and 23.7 ± 4.3 Tg yr^−1^ under the 30S_50R+50F_, 30S_100R+0F_, 50S_50R+50F_ and 50S_100R+0F_ scenarios, relative to the BAU, if the aim of potatoes as staple food policy achieved. These amounts are equal to about 0.7 ± 0.1, 1.4 ± 0.4, 1.7 ± 0.3 and 3.3 ± 0.9 million ha for early rice, based on a yield of 7.1 ± 1.3 Mg ha^−1^ of early rice, accounting for 12.1%, 24.1%, 29.3% and 56.9% of the total early rice sown (5.8 million ha) in China in 2012 [17]. It is predicted that the proportion of total water consumption used by agriculture would decrease to 50% by 2020, from the current 65% in some parts of Asia [47]. The water consumed by rice production accounts for 45% of total freshwater consumption in Asia [48], an amount that threatens continued high rice production in China. Switching to PSF could also decrease the early rice sown area by 12.1–56.9%, based on conventional levels of yield, by substituting different proportions of rice with potatoes. Hence, PSF have a large potential for mitigating the pressure on China’s rice production under the limited freshwater resources, ensuring China’s food security under future climate change.

Total GHG emission for rice, wheat and potatoes would be decreased by 3.4 ± 0.6 Gg CO_2_-eq yr^−1^ because the large reduction of GHG for rice in the BAU scenario. More 6.0 ± 0.8–13.0 ± 1.9 and 3.4 ± 0.4–19.9 ± 2.9 Tg CO_2_-eq yr^−1^ could reduce compared to BAU in 30% and 50% of PSF scenarios, respectively. The reductions of total GHG, equal to 1.1–9.0% of total CO_2_-eq emissions (319 Tg) associate with CH_4_ and N_2_O emitted from Chinese agroecosystems in 2005 [49]. Although net GHG of potatoes (3323 ± 1037–3636 ± 1140 kg CO_2_-eq ha^−1^) were significantly lower than that of rice and wheat in China (Figure 3), these values were significantly higher than that of 993–2350 kg CO_2_-eq ha^−1^ for potatoes in Iran and Portugal [50,51], which were caused by the higher CO_2_-eq emissions from N input and power use for irrigation in China. It was implied that China has a large potential to reduce total GHG emissions for staple food production by further reducing net GHG in potato production.

### 4.4. Effect of PSF on Potato Planting Under Different Scenarios

Total potato consumption would reach 17.4 ± 1.2 and 24.4 ± 2.0 Tg yr^−1^ in 30% and 50% PSF scenarios, respectively, in 2020. Under the yield level in 2012 (25.8 ± 7.9 Mg ha^−1^), the sowing area of potatoes were expected to reach 6.8 ± 2.2 and 9.5 ± 2.6 million ha in 30% and 50% PSF scenarios, which is close to our government’s target of increasing sown area of potatoes to 6.7 million ha by 2020 from 5.5 million ha in 2012, to achieve 30% of the PSF. To achieve the goal of 50% PSF, the planting area needs up to 10.0 million ha, while improving potato yields to more than 30 Mg ha^−1^, and no competition of land with rice, wheat or maize [52]. China has about 16.0 million ha of winter fallow cropland after late rice harvest in South China, and at least a quarter of this land can be used for planting potatoes, allowing a production of more than 60 million tons of food, enough to meet the food demand for 100 million people for 300 days [53]. In addition, 1.3 ± 0.4–3.3 ± 0.9 million ha for irrigated winter wheat in the NCP and/or early rice across China could be converted into potatoes, under 30–50% of PSF scenarios. Winter fallow cropland plus the potential substituted wheat and early rice by potatoes area was high to about 5.3–7.3 million ha, which is adequate for the additional demand of 1.2–4.5 million ha potatoes in China, while the average potato yield can be improved to 32.4 Mg ha^−1^ under optimized management practices (Appendix A). Adopting PSF could decrease the planting of rice and wheat, and using winter fallow land for potato production could ameliorate the shortage of arable land. Increasing the share of PSF may be an effective measure for mitigating the increasing pressure on producing more cereal grain production on limited arable land induced by rapidly urbanization [8,35], to ensure Chinese food security.

### 4.5. Effect of PSF on Food Energy and Nutrient Supply

The substitution of rice and flour with the same amount of potatoes will reduce the supply of plant food protein supply because of lower N content in potatoes than that in rice and flour [18,22]. However, this substitution has no significant effect on the total supply of plant food N, because only 3.1–7.5% of the total rice and flour consumption would be substituted by potatoes under different scenarios. According to data from China Food Composition, the per capita consumption of 141 g wheat flour, 150 g rice and 70 g rice flour per day together provide 63.1%, 53.5% and 92.6% of the recommended requirement of energy, macroelements and carbohydrates, respectively [54]; however, for many essential micronutrients, they provide about three-quarters to less than half of human bodily needs, and they provide none vitamins A and C [3]. The data from the national nutrition health survey results also showed that the Chinese overweight and obesity rates increased by 7.3% and 4.8% in 2010–2012 compared to 2002, and that there is still a deficiency of Ca, iron, vitamin A and vitamin D, and that the prevalence of chronic diseases such as hypertension and diabetes is growing rapidly and spreading to a younger group [16]. These alarming changes may have a close correlation with Chinese residents’ unhealthful staple-food consumption and nutrition structure. Potatoes, containing a variety of nutrients essential to the human body, include higher lysine content than wheat or rice. Furthermore, potatoes’ amino acid composition is close to that of easily absorbed soy protein, and they contain carotenoids not found in rice or wheat [54]. The potato flour can also make a significant contribution to the dietary intake of certain minerals including K, P, Mg and iron, as well as of dietary fibers, vitamin C and phenolic compounds [55], while delivering only modest amounts of saturated fatty acid and sodium [56]. If a resident who consumes 250 g of steamed bread per day as a staple food substituted 35% potato steamed bread for wheat-flour steamed bread, he could take 88.8% and 119.5% of the human daily requirement for vitamin C and K, far above the 19.6% and 42.8% of the daily vitamin C and K requirements from the same amount of ordinary steamed bread; and in addition, the total crude fat intake by a human body from potato steamed bread is lower than from ordinary steamed bread by about 21.8% [3] (Appendix A). Moreover, the total nutrient supply index of this study increased by 63.0%—from 0.27 for 100% wheat bread (250 g) to 0.44 for 35% potato bread (250 g; Appendix A), which represent potatoes as part of staple foods could supply more profitable nutrients than ordinary steamed bread. It was indicated that potatoes could increase the nutrient density of the diet by providing a relatively high micronutrient contribution, compared with energy content [56]. Our results also showed that the energy supply only 0.4–1.6% difference between potato substitution for rice and flour scenarios and only rice and flour scenario in 2012 (Appendix A). In other words, despite being from the same energy perspective of potato substitution for rice and flour, it will not significantly affect our results and conclusions. These suggest that PSF might improve the overall health of the Chinese people by improving dietary nutrition. However, the price of potato steamed bread is about double that of ordinary steamed bread in supermarkets [57]. The high price may result in the consumers unwilling to buy potatoes products. The Chinese government should not only support the development of advanced potato processing technologies and help increase the capacities of processing enterprises, but also subsidize processing enterprises to reduce the costs of potato based products [57], and to minimize the price gap between wheat-flour-only and potato-plus-wheat-flour products.

### 4.6. The Limitations and Future Research

We did a comprehensive environmental assessment of potato as staple food policy in China, focusing on the environmental benefits of substituting rice and wheat by potato. There are some limitations that the extrapolation of per capita habitual staple-food intake for 2013–2020 based on the historical trends from 2002 to 2012, using a simply common trend projection method [23], the possible effect of economic development on staple-food consumption should be considered in future research, and we predicted the amounts of potatoes consumed as a vegetable in urban and rural populations in 2020 by historical trends of tuber consumption from 2002 to 2012, and assuming that the proportion of potatoes to other tubers keep constant as reported in 2012 [5], because of no available data on the future condition. Moreover, we designed different proportions of potato substitutes for rice and flour and substituting potatoes for equal amounts of rice and/or flour, and we illustrated that this substitution had no significant effect on food energy supply from rice, flour plus potatoes in all substitution scenarios, future research can be tried to use equal amounts of the nutrition perspective to substitute for rice and flour with potatoes. The N, P_2_O_5_ and K_2_O sourced from organic fertilizers were excluded in this study because of the reasons described in the data collection section, this part of N-, P_2_O_5_- and K_2_O-fertilizers should be included in the comparison of N use efficiencies of three crops in future research. We did not consider the potential effect of PSF on the types and prices of staple foods. The price of potato steamed bread is about double that of ordinary steamed bread in supermarkets [57], because until now the price of potato granules is about triple the price of wheat flour because the processing industry is of a low capacity in China. The high price may result in the consumers unwilling to buy potatoes products. How to minimize the price gap between wheat-flour-only and potato-plus-wheat-flour or rice flour products is urgently needed for policy makers. Moreover, the implementation of PSF might generate certain effects on the inputs and outputs of potatoes, rice and wheat production, the price of associate food products, and cost of some economic input for establishing the corresponding processing industry and for promoting this policy, hence, the cost–benefit analysis of potato production to the rest of selected cropping systems in term of the environmental impact assessment, and the cost of CO_2_-eq emissions of the selected cropping systems compared to the potato system is urgently worth doing in future research.

## 5. Conclusions

In this study we studied the comprehensive effect of the government’s target of having 30% of total consumed potatoes as a staple food in China, on chemical N-, P_2_O_5_- and K_2_O-fertilizer, irrigation-water inputs and total GHG emissions for rice, wheat and potatoes and on energy supply and nutrition supplies, by substituting potatoes for equal amounts of rice and/or flour. The results showed that 30% of the consumed potatoes as a staple-food policy could decrease total chemical N fertilizer and irrigation water inputs, reduce total GHG emissions and mitigate P_2_O_5_ fertilizer inputs for three crops, relative to the 2012 and business as usual scenario, by reducing early rice across China, and/or winter wheat in the NCP. The decreases in irrigation water in different substitute scenarios compared to 2012 are equivalent to the water use of about 17.9 ± 4.9–21.8 ± 4.8 million people in 2015. The reduction in winter wheat production in the NCP could convert 6.1–11.4% of the typical winter wheat–summer maize double cropping system to a potato–summer maize system and a winter wheat–summer maize–spring maize system, with three harvests every two years, to mitigate the decline of the groundwater table and a 12.1–24.1% of reduction in early rice sowing could mitigate the pressure on the increasing in rice demand anticipated by China’s increasing population under limited freshwater resources in the future. More N- and P_2_O_5_-fertilizer, irrigation-water and total GHG could be reduced if the proportion of PSF rose to 50%, coupled with improving potato yields. Our results implied that China can take the challenges of producing more grains anticipated by population and economic growth with fewer inputs, and with reduced environmental costs by making potatoes entered into staple diets, while ensuring an equal amount of energy supply and more healthful nutrition supplies. In addition, potato as a staple food policy might contribute to the adjustment of agricultural structure in some areas with serious water deficits under climate change, especially irrigated wheat on the NCP.

## Figures and Tables

**Figure 1 ijerph-16-02700-f001:**
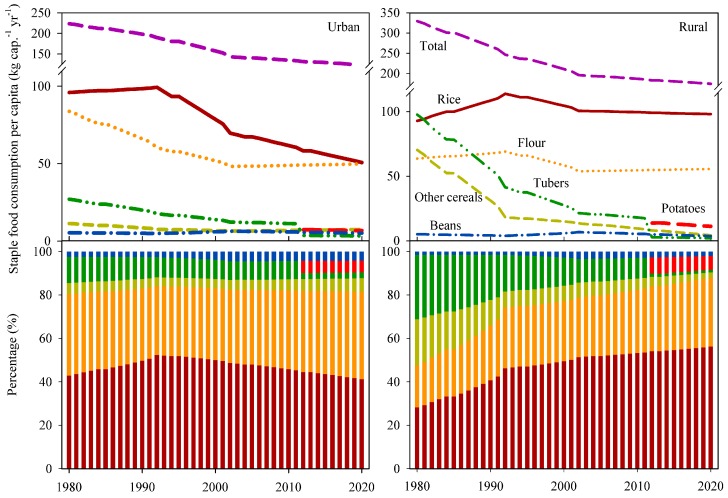
Staple-food consumption per capita and proportions in Chinese urban and rural households during 1980–2012, and projected future trends to 2020.

**Figure 2 ijerph-16-02700-f002:**
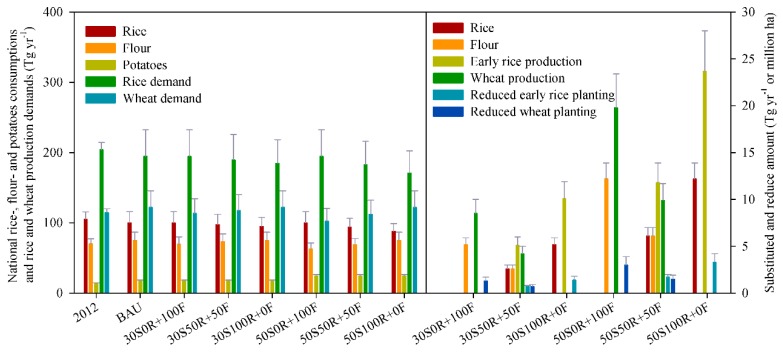
National rice-, flour- and potato consumptions; rice and wheat production demands (Tg yr^−1^; **left**); potatoes as staple foods substituted for rice and flour, and rice and wheat production (Tg yr^−1^); reduced early rice and wheat planting (million ha) under the different substitution scenarios in 2020, in comparison with 2012 (**right**).

**Figure 3 ijerph-16-02700-f003:**
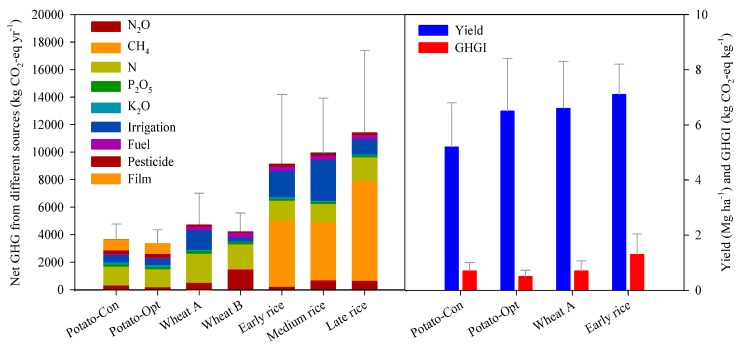
Total GHG emission from different CO_2_-eq emissions in conventionally grown (Potato-Con), and optimally grown (Potato-Opt) potatoes, winter wheat in the North China Plain (NCP; wheat A), winter wheat across China except for NCP (wheat B) and early, medium and late rice in China (**left**) and the yields and GHGI of Potato-Con, Potato-Opt, wheat A and early rice (**right**).

**Figure 4 ijerph-16-02700-f004:**
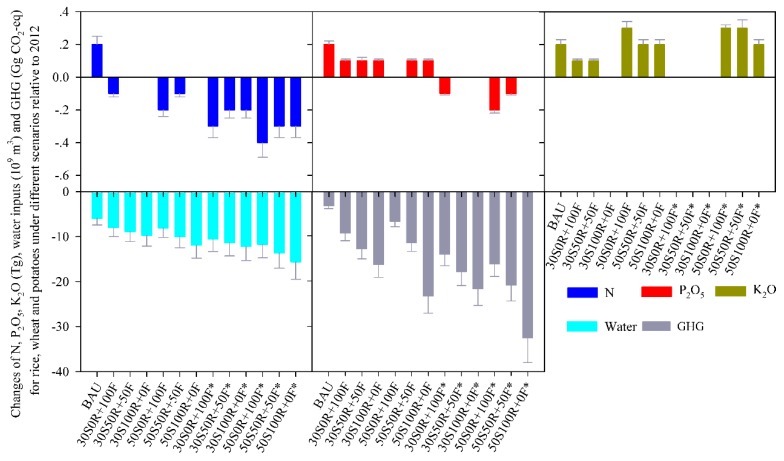
Chemical N-, P_2_O_5_- and K_2_O-feitilizer inputs, irrigation-water consumptions and total GHG for rice, wheat, conventionally grown potatoes (Potato-Con), and optimally grown potatoes (Potato-Opt) under different scenarios, compared with 2012.

**Table 1 ijerph-16-02700-t001:** Potatoes-as-a-staple-food (PSF) ratios under different scenarios, with different proportions of potato substituted for rice and flour in China in 2020.

Code	PSF Ratio	Substituted for Rice	Substituted for Flour
BAU	30%	0	0
30% Scenarios			
30S_0R+100F_	30%	0	100%
30S_50R+50F_	30%	50%	50%
30S_100R+0F_	30%	100%	0
50% Scenarios			
50S_0R+100F_	50%	0	100%
50S_50R+50F_	50%	50%	50%
50S_100R+0F_	50%	100%	0

BAU: business as usual.

**Table 2 ijerph-16-02700-t002:** Required per capita potato-as-vegetable (VP) and potato-as-a-staple food (PSF) consumptions, under the goals of 30% and 50% of PSF consumption in China’s urban and rural areas in 2020.

Area	Ratio of PSF/(VP+PSF)	VP	PSF	VP + PSF
		kg cap.^−1^ yr^−1^		
Urban	30%	6.7 ± 1.0	2.9 ± 0.4	9.6 ± 1.1
	50%	6.7 ± 1.0	6.7 ± 1.0	13.5 ± 1.4
Rural	30%	11.0 ± 1.7	4.7 ± 0.7	15.7 ± 1.8
	50%	11.0 ± 1.7	11.0 ± 1.7	22.1 ± 2.3

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
