# Peer review of "Comprehensive Environmental Assessment of Potato as Staple Food Policy in China"

_ijerph, 2019, doi:10.3390/ijerph16152700_

Round 1
Reviewer 1 Report
This Study analyses potato production and implications versus environmental challenges. I suggest to the author to provide the emission per selected systems to that of potato instead of given the total emission. Moreover, I was expecting to see the cost-benefit analysis of potato production to the rest of selected farming systems in term of environmental impact assessment. It is true that the study provides potential emission of CO2eq therefore, there is a need to evaluate the cost of that emission of each selected crop and farming systems compared to the potato.
The study is relevant and worth for publication.after this minor correction.
Author Response
This study analyses potato production and implications versus environmental challenges. I suggest to the author to provide the emission per selected systems to that of potato instead of given the total emission. Moreover, I was expecting to see the cost-benefit analysis of potato production to the rest of selected farming systems in term of environmental impact assessment. It is true that the study provides potential emission of CO2eq therefore, there is a need to evaluate the cost of that emission of each selected crop and farming systems compared to the potato.
The study is relevant and worth for publication after this minor correction.
We compared the per unit area emissions between potato system and the selected systems in this study (Fig. 3), and estimated the total emission under different proportions of potato substitutes for rice and flour, to show the potential reduction of total emission through comparing with total emission under business as usual scenario (Fig. 4). In this study we mainly focus on the environmental impact assessment of different proportions of potato substitutes for rice and flour, by assuming that the potato as staple food policy could be achieved in near future, and neglected the cost-benefit analysis of potato production to the rest of selected cropping systems in term of environmental impact assessment, and the cost of CO2eq emission of the selected cropping systems compared to potato system, because of there no available information on the detailed cost of this conversion. We have added a sentence to discuss this deficiency in the discussion section, as lines 642-648 in the added discussion section 4.6
.
Reviewer 2 Report
The paper called “Comprehensive environmental assessment of potato as staple food policy in China” assess the environmental impact of potato as staple food in China, focusing in the environmental benefits of substituting rice and wheat by potato.
· L21 – Name of chemical formulas should be introduced: “N”, “P2O5”, “K2O”.
· L24 – When you say this “PSF consumed are expected to reach 5.2 Tg” is this per year?
· Keywords: In my opinion you do not need “Potato staple-food” because it is already in the paper title.
· L59 and 87 – Repeated information in both places in the paper.
· L137 – Please be more precise in the terminology throughout the paper, e.g. “hidden CO2 from”, i.e. indirect emissions.
· L154 – What is the basis to assume that the intake partition between 2013-2020 will be the same as 1980-2012?
· L157 – Insert a space between “were” and “7.7”.
· L210 – Figure 4 is not cited in the text. I think it is relevant, but it needs to be moved into the results section, cited and explained.
· L228 – Figure 1 is not is not a result of this paper, thus I recommend remove it or move to SI file.
· Section 2.1 – What is the source/share (inorganic and organic) for fertilizer considered? This might be relevant for the results that you are obtaining. You should consider this in your work, namely considering different scenarios for fertilizer sources.
· I found multiple typos throughout the manuscript. Please revised it carefully.
· It was not clear to if the N, P2O5 and K2O inputs were proportional to the area or to the production. This is particularly relevant when you consider changes in the future yields.
· Results are well presented in the “Results” section, however I would like to have an uncertainty assessment of results. This would demonstrate the robustness of obtained results.
· In the “Discussion” section, it should be included a sub-section regarding the limitations and possible improvements of the presented work.
Author Response
Reviewer 2:
The paper called “Comprehensive environmental assessment of potato as staple food policy in China” assess the environmental impact of potato as staple food in China, focusing in the environmental benefits of substituting rice and wheat by potato.
L21 –Name of chemical formulas should be introduced: “N”, “P2O5”, “K2O”.
We have added the name of chemical formulas for N, P2O5 and K2O.
L24 –When you say this “PSF consumed are expected to reach 5.2 Tg” is this per year?
Yes, we have added the unit of yr-1 for the associate contents.
Keywords: In my opinion you do not need “Potato staple-food” because it is already in the paper title.
We deleted it as the reviewers’ comment and added ‘Nutrient Reference Values’ as one of the keywords.
L59 and 87–Repeated information in both places in the paper.
We deleted the repeated sentence of “if potato staple-food products could be widely accepted by the public” to avoid repeat.
L137 –Please be more precise in the terminology throughout the paper, e.g. “hidden CO2 from”, i.e. indirect emissions.
We revised the hidden CO2 to indirect emissions as reviewers’ comment.
L154–What is the basis to assume that the intake partition between 2013-2020 will be the same as 1980-2012?
We extrapolated per capita habitual staple-food intake for 2013-2020 based on the historical trends from 2002 to 2012, using a simply common trend projection method, we added a sentence and an associate publication for explain it, as lines 172-173.
L157–Insert a space between “were” and “7.7”.
We have done.
L210–Figure 4 is not cited in the text. I think it is relevant, but it needs to be moved into the results section, cited and explained.
We have revised the manuscript as the reviewers’ comment.
L228–Figure 1 is not is not a result of this paper, thus I recommend remove it or move to SI file.
We moved it to SI file as the reviewers’ suggestion.
Section 2.1 – What is the source/share (inorganic and organic) for fertilizer considered? This might be relevant for the results that you are obtaining. You should consider this in your work, namely considering different scenarios for fertilizer sources.
The amounts of N, P2O5 and K2O from organic fertilizer were not direct reported in the selected literature as that of chemical fertilizers, the ratio of organic fertilizers application area to total sown area is relatively low, especially for rice and wheat production, and they can’t be easy to calculate because the complex types of organic fertilizers and no available data on the contents of N, P2O5 and K2O in some organic fertilizers. In addition, the organic N-, P2O5- and K2O-fertilizers are belong to the internal recycled resources in the food production-consumption system (Ma et al., 2012; Gao et al., 2019), and we mainly emphasis on the new resources input to food system and want to evaluate the contribution of PSF to national goal of zero growth in fertilizer use, hence, the N, P2O5 and K2O sourced from organic fertilizers were excluded in this study. We added a sentence for explain this in MM section, as lines 137 to 147.
I found multiple typos throughout the manuscript. Please revised it carefully.
We have modified the manuscript carefully.
It was not clear to if the N, P2O5 and K2O inputs were proportional to the area or to the production. This is particularly relevant when you consider changes in the future yields.
The production efficiencies of chemical N-, P2O5- and K2O-fertilizers and irrigation water, and GHG emissions of rice, wheat and potatoes were the mean levels of the collected data in this study (Table S1–S5), which can represent the mean pattern of farmer’ practices in the last three decades and the near future as much as possible, hence we assumed that these efficiencies and GHGI of three crops no variation in all scenarios. Under this condition, the N, P2O5 and K2O inputs were proportional to the production. We added a paragraph for describe it in MM section, see lines 197 to199.
Results are well presented in the “Results” section, however I would like to have an uncertainty assessment of results. This would demonstrate the robustness of obtained results.
We did uncertainty assessment of results using the error propagation equation of mathematical statistics (IPCC, 2001) by calculating or assuming the percentage uncertainties of the collected data from different sources, as described in MM section 2.5 and we add a paragraph for describing of the error propagation equation of mathematical statistics. And added SD values for the associated results appeared in the manuscript, figures and tables to show the uncertainties of our results, as the revised table 2, figure 3.
In the “Discussion” section, it should be included a sub-section regarding the limitations and possible improvements of the presented work.
We have added a section of 4.6 to show the limitations of this study and future research, as lines 618 to 648 in the revised manuscript.
Reviewer 3 Report
- Revise the article for typos
- The order of Figures should be insequence (i.e. first Figure 1, then 2 in the text etc).
Author Response
Reviewer 3:
Revise the article for typos
The order of Figures should be insequence (i.e. first Figure 1, then 2 in the text etc).
We have modified the manuscript carefully and revised the order of figures, and hope to improve it as much as possible.